# Effectiveness of Custom-Made Foot Orthoses vs. Heel-Lifts in Children with Calcaneal Apophysitis (Sever’s Disease): A CONSORT-Compliant Randomized Trial

**DOI:** 10.3390/children8110963

**Published:** 2021-10-25

**Authors:** Javier Alfaro-Santafé, Antonio Gómez-Bernal, Carla Lanuza-Cerzócimo, José-Víctor Alfaro-Santafé, Aitor Pérez-Morcillo, Alejandro-Jesús Almenar-Arasanz

**Affiliations:** 1Department of Podiatry, Faculty of Health Sciences, Manresa University, 08242 Manresa, Spain; javieralfaro@podoactiva.com (J.A.-S.); carlalanuza@podoactiva.com (C.L.-C.); victoralfaro@podoactiva.com (J.-V.A.-S.); aitorperez@ucam.com (A.P.-M.); 2R & D Department, Biomechanical Unit, Podoactiva Headquarters, 22197 Huesca, Spain; aalmenar@usj.es; 3Departament of Podiatry, Faculty of Health Science, Campus de Los Jerónimos, Universidad Católica San Antonio de Murcia, Guadalupe, 30107 Murcia, Spain; 4Department of Physiotherapy, Faculty of Health and Sports Sciences, San Jorge University, 50830 Villanueva de Gállego, Spain

**Keywords:** calcaneal apophysitis, children, Sever’s disease, treatment, orthoses

## Abstract

The aim of the present research was to determine the effectiveness of relieving calcaneal apophysitis pain using “off-the-shelf” heel-lifts and custom-made orthotics. Two intervention modalities were evaluated and compared in a 12-week follow-up trial. Inclusion criteria included 9- to 12-year-old children diagnosed with calcaneal apophysitis. Children were randomly stratified into treatment A (custom-made polypropylene foot orthoses) and treatment B (“off-the-shelf” heel-lifts) groups. Treatment effectiveness was measured by algometry and the visual analogical scale (VAS). A total of 208 patients were included. The treatment A group showed an increase in threshold algometry of 53.4% (95% CI 47.1% to 59.7%) and a decrease in VAS of −68.6% (95% CI −74.5% to −62.7%) compared with the treatment B group (*p* < 0.001). Calcaneal apophysitis pain perception was improved in both groups, but children who used custom-made foot orthoses showed a greater improvement.

## 1. Introduction

Calcaneal apophysitis (Sever´s disease) is a very common ailment present in the heels of children between 7 and 15 years old [1,2,3,4,5]. The literature describes calcaneal apophysitis as a syndrome caused by overuse, resulting from the production of repetitive micro-traumas [3]. The mechanical etiology relates the injury to the traction forces of triceps surae and plantar fascia on the calcaneus bony surface [3,4,5,6]. The anatomical unit named the “Achilles–calcaneus–plantar System” (ACPS) describes the functional connection among the Achilles tendon, the calcaneus bone, and the plantar fascia [6]. There, the posterior trabecular calcaneal system works as a sesamoid bone between the fascia and the tendon fibers [6]. Therefore, repetitive stress of the Achilles tendon and the plantar fascia is transmitted onto the calcaneus surface affecting bone remodeling, creating perpendicular fibrous bands of cartilage in the secondary ossification center of the calcaneus [7]. This, in addition to repetitive impacts on the bony surface, constitutes the focus of calcaneal apophysitis [5,6,7,8].

Calcaneal apophysitis, fortunately, is a benign ailment which disappears without exception after puberty when the secondary ossification center of the calcaneus is closed [9]. Unfortunately, currently, no evidence-based treatments are available [10]. Nevertheless, conservative ones are the most commonly used; these include sport activity modification, stretching, and strengthening exercises and the application of podiatric strategies (heel-lifts and foot orthoses) [5,9,10,11,12].

The described conservative strategies are popular [10] but present a number of drawbacks. Activity cessation means an increase in sedentary lifestyle, which is a major factor in obesity [13]. The prescription of “off-the-shelf” heel-lifts is very widespread [9,10,11,12]. These reduce pain perception due to the elevation of the heel, shortening the distance between the origin of the triceps surae muscle and its insertion on the calcaneus [12]. Unfortunately, heel-lifts do not act completely on the mechanical etiology of calcaneal apophysitis [3,6,8].

Along this line, there are some studies in the literature that have used orthoses to act both on ACPS and repetitive impacts [8,11,12,14,15,16]. Orthoses provide support in the medial arch (relaxing plantar fascia), including a heel-lift component and a wider support surface (dispelling impacts) [8,11,12,14,15,16]. The available evidence reports the use of orthoses [8,11,14], but most of them were prefabricated and the studies revealed statistical shortcomings and methodological concerns that limit the validity of the reported results [8,16]. No studies have previously compared the use of “off-the-shelf” heel-lifts with custom-made orthotics [16]. Based on this, the present research aimed to provide a pragmatic randomized comparative effectiveness trial with an intervention period of 12 weeks, with the aim to compare heel pain perception in children with calcaneal apophysitis using custom-made polypropylene foot orthoses and “off-the-shelf” heel-lifts. Therefore, it was hypothesized that the primary outcome, pain relief, would be significantly improved with the custom-made orthosis compared to the heel-lift.

## 2. Materials and Methods

The study was a parallel-group, randomized comparative effectiveness trial with concealed allocation, blinding of investigators and assessors, and intention-to-treat analysis. It examined the effect of custom-made foot orthoses and heel-lifts in children with calcaneal apophysitis. Participants were enrolled at the time of calcaneal apophysitis diagnosis. After baseline assessment, children were individually randomized to “custom-made foot orthoses” (treatment A group) or “off-the-shelf heel-lifts” (treatment B group). Concealed allocation was carried out by having randomization performed by a third party who was not involved in the recruitment or treatment of the children. Eight permuted blocks were used to stratify randomization by body mass index (BMI) (19 or lower, versus > 19 kg/m^2^), lunge test (32° or lower, versus > 32° of dorsal flexion), foot posture index (FPI-6) (4 or lower, versus > 4), and visual analogical scale (VAS)score (74 or lower, versus > 74 mm) as these are considered to be important risk factors associated with pain severity in calcaneal apophysitis [17,18,19]. A researcher who was unaware of the randomized group allocation measured the outcomes at baseline and 12 weeks later. The design of the present investigation was based on and executed according to the CONSORT Statement. The study was approved in 2017 by the Ethics Committee of Clinical Research of Aragón (C.P.-C.I.PI16/0303) and registered at ClinicalTrials.gov (NCT03960086).

### 2.1. Participants

All children with calcaneal apophysitis diagnosis who came for an orthopedic treatment at a Podiatric Clinic between June 2019 and February 2020 were assessed for study eligibility prior to treatment application. The inclusion criteria included boys and girls physically active between 9 and 12 years old diagnosed radiologically with calcaneal osteochondritis in one foot [12]. This age group was chosen because it corresponds to the mean age where calcaneal apophysitis is presented for many authors [1,5,11]. Children were excluded if they had suffered some trauma on the heel in the past 2 months; had received anti-inflammatory drugs and/or physical treatment for pain in the past 3 months; had presented physical or neurological impairment; were sedentary children; or were not interested. Informed consent forms with parents´ and children’s authorization were required as part of the study.

### 2.2. Interventions

Before treatment application, all children received the following information about conservative strategies for pain reduction: triceps surae stretching, sport activity intensity reduction but no cessation, and 10 min of ice application for pain exacerbation [10].

Each intervention (custom-made foot orthoses and heel-lifts) consisted of a treatment period of 12 weeks, following the design of previous studies [10]. Both interventions were prescribed, designed, and fabricated by a podiatrist expert in orthopedics, who was unaware of the randomized group allocations. Completely blinding of the podiatrist was carried out as he designed both treatments for each patient regardless of group allocation; therefore, he did not know if the children had ultimately received custom-made foot orthoses or heel-lifts.

#### 2.2.1. Treatment A Group (TA)

Children in the treatment A group received custom-made foot orthoses as treatment intervention. The orthoses had a thickness of 9 mm and were composed of the following materials: confortene, polypropylene, poron XRD, and lunasoft (Figure 1). These kinds of orthoses have been previously used in the literature [20]. Children were advised pragmatically to wear the orthoses at least 8 to 10 h per day and during sport activity.

#### 2.2.2. Treatment B Group (TB)

Children in the treatment B group received an 8 mm “off-the-shelf” heel-lift as treatment intervention. These were composed of confortene, poron XRD, and ethyl vinyl acetate (EVA) (Figure 2). Children were advised pragmatically to wear the heel-lifts at least 8 to 10 h per day and during sport activity.

Children were blinded to group allocation. Both treatment A and B groups received the same attention during the consultation, and they were informed that the intervention given was adequate for their disease before treatment application and after the follow up.

### 2.3. Outcome Measures

The primary outcome and the characteristics of the sample of the present research were obtained at baseline and after 12 weeks of follow up by an experienced and trained assessor who was blinded to group allocation. At baseline, participants’ weight and height were measured with an Año-Sayol scale and stadiometer, respectively (Año-Sayol SL, Barcelona, Spain).

The primary outcome was calcaneal apophysitis pain perception, which was determined by the VAS and the algometry threshold. The VAS is commonly used to measure pain perception [21] and has been used before to measure pain perception in children with calcaneal apophysitis [14,18]. The VAS is described as a horizontal line of 100 mm in length, where at each end point, the words ‘‘No pain’’ and ‘‘Worst imaginable pain” are placed [21,22]. Participants were asked to mark the VAS line at the point which best represented their pain intensity [19]. Algometry is used to measure pressure pain threshold [23]. It is a technique that has previously shown an excellent intrarater reliability with intraclass correlation coefficient (ICC = 0.91) [24]. To perform the algometry, the Wagner FPXTM 25 Algometer (Wagner Instruments^®^, Greenwich, CT 06836-1217 USA) was used. It has been used before in children with calcaneal apophysitis [10]. To perform the algometry, children were asked to lie down in prone position on the stretcher, with the knee and ankle bent 90°. Then, the algometer was positioned on the Achilles tendon insertion on the calcaneus. Three separate measurements were obtained and the average of them was taken as the algometry value [10].

Values for the BMI, FPI-6, and lunge test in the children with calcaneal apophysitis were taken as the baseline characteristics of the sample. The FPI-6 was performed following the guidelines of Redmond et al. [25]. FPI is a 6-point tool for clinical assessment, which evaluates the multisegmented nature of foot posture in the three spatial dimensions. Each component of the test is graded between −2 and +2 (signs of supination or pronation), where neutral is graded with 0. Finally, when the score ranges from 0 to 5, the foot is considered as normal; pronated when it ranges from 6 to 9; highly pronated when >10; supinated when it ranges from −1 to −4; and highly supinated when a score from −5 to −12 is obtained [25,26,27]. The lunge test is commonly used to determine ankle dorsiflexion restriction and shortness in the triceps surae muscle [2,27]. To perform the lunge test, children were asked to stand and move one foot backward into a comfortable position. Then, they were asked to perform a forward ankle dorsiflexion that was measured with the Tiltmeter mobile application (IntegraSoftHN—Carlos E. Hernández Pérez). Restriction in the length of this muscle is considered when the dorsal flexion of the ankle is less than 32° [17].

### 2.4. Data Analysis

Sample size estimation was based on the detection of a difference of 29% in the number of children with pain and a mean difference of 0.39 or higher for the algometry and the VAS between baseline and after the follow up based on a test of a hypothesis comparing proportions between two related groups. Assuming a two-sided α of 0.05, power of 80%, and a 10% drop-out rate, a sample size of 198 participants was required. Continuous variables were expressed as the mean ± standard deviation (SD), whereas qualitative variables were expressed as frequencies and percentages. Continuous data were checked for normality by the Kolmogorov–Smirnov test.

The baseline characteristics of the children were summarized using descriptive statistics and tabulated for comparison between both groups. Baseline characteristics were compared between included and excluded participants to analyze whether they were representative. Mann–Whitney U and Student’s t-tests were used to compare continuous variables, and the Fisher Test was used to compare dichotomous variables.

Differences between baseline and final assessment in algometry and VAS were performed with the calculation of the “percentage change”. Positive results indicated an increase in values at the end of the study, and negative results indicated a decrease. Treatment effectiveness after the follow up was determined by regression models. Linear regression was applied to quantitative variables, algometry, and VAS, and logistic regression was applied to dichotomous qualitative variables; association was measured in terms of odds ratio.

Statistical analysis of the outcomes was performed according to intention-to-treat principles, comparing the groups regarding pain perception evolution. For all tests, a two-sided *p*-value < 0.05 was considered significant. The statistical analyses were performed using the SPSS software 22.0 for Windows (SPSS Ibérica, Madrid, Spain).

## 3. Results

### 3.1. Enrollment and Characteristics of the Participants

Between June 2019 and February 2020, 234 children were screened, 208 of which were included in the present study. Among these, 104 children were allocated to both treatment A and B groups. The flow of participants throughout the study is presented in Figure 3, and baseline characteristics are presented in Table 1. No significant demographic differences were found between the two groups. BMI values suggested that children in both groups were of normal weight; lunge test values were accurate for this age group; and FPI-6 values were greater than eight (>4) in both groups, respectively. Results showed that children had, respectively, shortened triceps surae muscles and pronated feet.

### 3.2. Compliance with Study Protocol

Children had a reasonable compliance with their allocated interventions. Only four and five children were lost to follow up in the TA and TB groups, respectively (Figure 3). These children withdrew from the study or were not reachable.

### 3.3. Outcomes

The primary outcome of calcaneal apophysitis pain perception analysis is shown in Table 2. Larger changes and improvements were noted for the groups where participants used custom-made foot orthoses. Differences between baseline and final assessment were statistically significant (*p* < 0.05) for both interventions. The values for VAS decreased by 68.5 ± 15.4 points in the treatment A group (*p* < 0.001) and 14 ± 17.7 points in the treatment B group (*p* < 0.001); the algometry values increased by 2.0 ± 0.5 kgf in the treatment A group (*p* < 0.001) and 0.6 ± 0.6 kgf in the treatment B group (*p* < 0.001). Differences in changes and improvement between treatment A and B groups are shown in Figure 4 and were statistically significant for all the variables (*p* < 0.001). Results were compared between groups using odds ratio, and the confidence intervals were wide, as shown in Table 2.

The odds ratio (95% CI) between groups showed that children who wore custom-made foot orthoses had a higher improvement, which increased algometry data by 53.4 (47.1 to 59.7) and reduced VAS by 68.6 (74.5 to 62.7), compared with children who wore heel-lifts.

## 4. Discussion

The purpose of this trial was to compare heel pain perception in children with calcaneal apophysitis using custom-made polypropylene foot orthoses and “off-the-shelf” heel-lifts in an intervention period of 12 weeks. Calcaneal apophysitis pain perception for the three variables measured by VA and algometry were significantly improved and reduced in both groups. The treatment A group showed significant pain relief compared with the treatment B group.

At baseline, all the participants had high VAS values and a reduced pressure pain threshold on the affected heel. Pain relief was significantly different between treatment A (custom-made foot orthoses) and treatment B (heel-lifts) groups.

The heel-lift’s function was to lift the heel with an inclined plane, which allowed a reduction in Achilles tendon tension and traction on the calcaneus bony surface [3,8,9,10]. On the other hand, custom-made foot orthoses provided a lift–rise component in the heel; an increased support surface covering the calcaneus plantar face, reducing repetitive impacts; and a pronation correction component tailored to the foot of each child [3,8,10].

Improvement in the treatment B group was found in approximately 20–30% of children, while in the treatment A group, it was found in 70–90% of children (*p* < 0.001). Compared with the treatment B group, the treatment A group experienced an increase in the algometry threshold of 53.4% and a VAS punctuation reduction of −68.6%. Similar results were obtained in 2011 in two studies performed by Perhamre et al. [8,9]. In their research, the authors compared a heel-cup (3 mm), which reduced repetitive impacts with a wedge that lifted the heel (5 mm) in 51 boys with calcaneal apophysitis; the cup produced pain reduction by 80%, due to its higher impact absorption. They employed the Borg CR-10 visual analogue scale, obtaining a significant decrease in pain levels from 7 to 2. Between 2010 and 2016, James et al. [14] performed a randomized controlled trial where they compared the effectiveness of a heel-lift (6 mm EVA) with a prefabricated foot orthosis (polyurethane). At the beginning, their study was planned to last 3 months but eventually lasted 12 months. Pain intensity was measured by the “Faces pain scale”, not obtaining differences between the two treatment tools in the 12-month follow up. In the present study, we saw significant differences in 12-week intervention periods. Calcaneal apophysitis is considered a disease of growth age, which will disappear at the end of the calcaneus ossification, i.e., long-term monitoring results may not be significant [9,10]. James et al. [14] also did not employ an individualized treatment, while they applied prefabricated foot orthoses. In our case, custom-made foot orthoses were individually adapted, as recommended by Landorf et al. [28].

Another interesting finding was that children in both groups presented a BMI similar to that found by James et al. in children with calcaneal apophysitis [18] and presented flat feet according to FPI-6 (>8 in both groups) [27]. In the current literature, we can find several studies in which authors have analyzed the relationship among FPI-6, calcaneal apophysitis, weight, and age [18,26,29,30,31]. In 2015, Evans and Karimi [31] analyzed the relationship between BMI and FPI-6 in 728 overweight and healthy children between 3 and 15 years of age; they did not find a significant association between BMI and flat feet. Gijon-Nogueron et al. [30] evaluated in a cross-sectional study 1762 school children between 6 and 11 years of age, without pain and/or injury in the feet and lower limbs. Results showed the generally accepted margins of neutral FPI-6 (0 to 4). Martínez-Nova et al. [29] supported the FPI-6 results provided by Gijon-Nogueron et al. [30] in healthy children. In another study performed by James et al. [18], they recruited 124 children with calcaneal apophysitis between 8 and 14 years of age. The authors found that children had a higher BMI and FPI-6 values compared to population norms [18], while according to the authors, the ankle range of motion was increased. In the same line, Hawke et al. [26] found a relationship between flat feet and ankle dorsiflexion limitation assessed by the lunge test in 30 healthy children between 7 and 15 years of age. Our sample was composed exclusively of children with calcaneal apophysitis who presented flat feet and a higher BMI compared with population norms [18]. In their research, James et al. [18] observed that these were risk factors associated with calcaneal apophysitis pain. In his research, Huerta [6] showed the relationship between the triceps surae muscle and plantar fascia, and how the tightness in the muscle increases Achilles tendon tension, which is reflected as ankle dorsiflexion stiffness and plantar fascia tension during weight-bearing activities.

Our findings suggest the children with calcaneal apophysitis of the present research have higher BMIs compared with population norms and flat feet [2,18,25], but no ankle dorsiflexion restriction according to the normative reference values provided by McKay et al. [17]. Heel-lifts of the treatment B group acted exclusively on Achilles tendon tension, which was normal in our research [17]. Therefore, heel-lifts are an insufficient approach for calcaneal apophysitis. On the other hand, custom-made foot orthosis included a wider surface, dismissing the repetitive impacts and a pronation correction, which reduced the stress on the plantar fascia [6,8].

Alongside the results found, some limitations need to be considered. First, children in both groups wore their respective treatment in their own footwear, rather than in a standardized shoe. Second, the level of physical activity of each participant was not considered. Third, the follow-up period lasted 3 months; therefore, changes in short- or long-term periods were not investigated. Fourth, no physical exercise examination tests were performed in order to determine how real sport activity could influence the presence of pain. The present study provides new information about approaches to calcaneal apophysitis. The strengths of this study are the following: participants and assessors were blinded, the sample size was sufficient to show reliable results, stratified randomization was performed in eight permuted blocks considered as important risk factors in calcaneal apophysitis pain, both groups (treatment A and B) were homogeneous, and the study provided consistent data about the use of custom-made foot orthoses for calcaneal apophysitis pain relief.

## 5. Conclusions

The potential use of custom-made foot orthoses in the approach to calcaneal apophysitis needs to be considered in clinical consultation as an effective treatment strategy. Future studies should consider the comparison between custom-made foot orthoses against other treatment strategies and protocols, such as strengthening and stretching exercises, physiotherapy approaches, drug treatment, and physical exercise tests, for the evaluation of pain perception.

In summary, this study highlights that the use of custom-made foot orthoses instead of heel-lifts for calcaneal apophysitis (Sever´s disease), applied during a 12-week follow-up period, may have a substantial effect on calcaneal apophysitis pain relief.

## Figures and Tables

**Figure 1 children-08-00963-f001:**
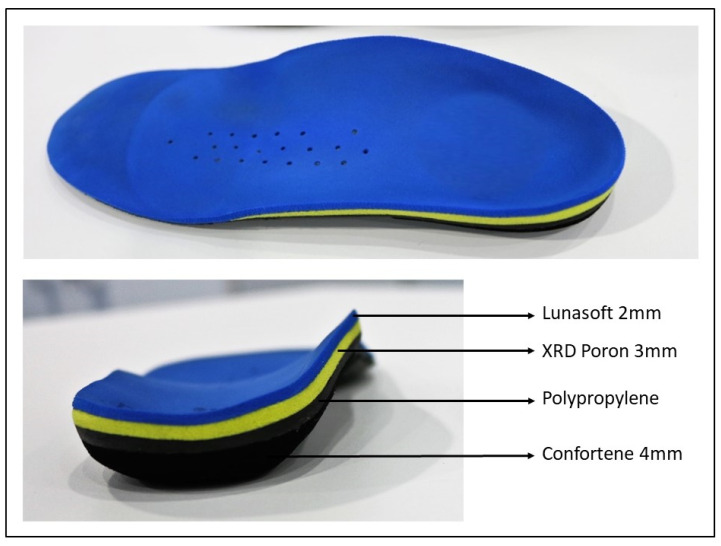
Custom-made polypropylene foot orthosis components.

**Figure 2 children-08-00963-f002:**
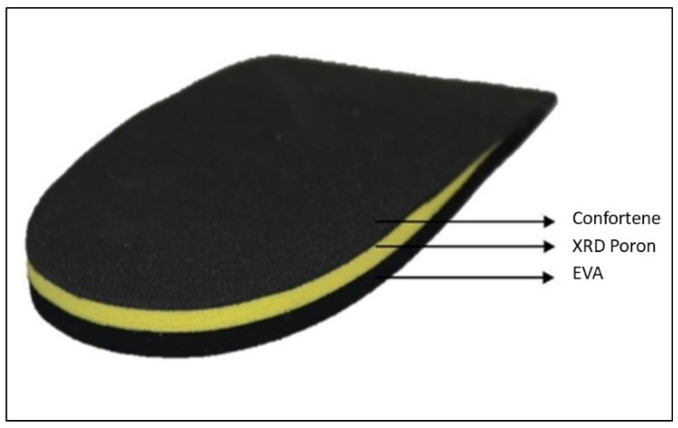
Off-the-shelf heel-lift components. EVA: ethyl vinyl acetate.

**Figure 3 children-08-00963-f003:**
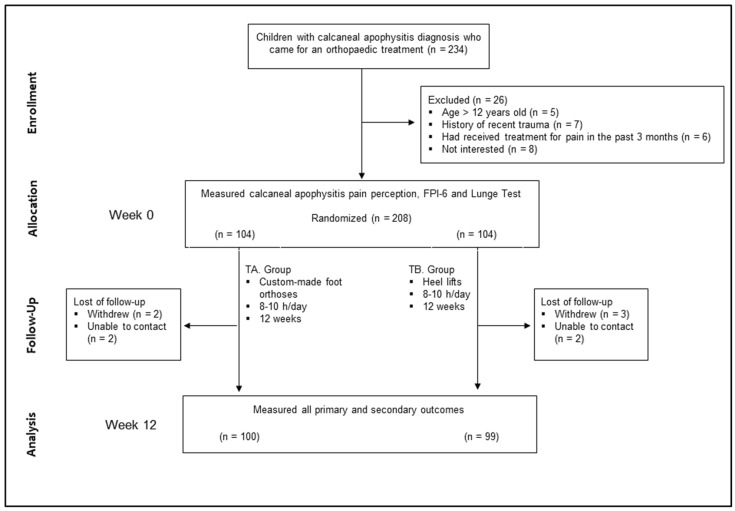
Design and flow of the children through the trial. TA = treatment A group; TB = treatment B group; FPI = Foot Posture Index.

**Figure 4 children-08-00963-f004:**
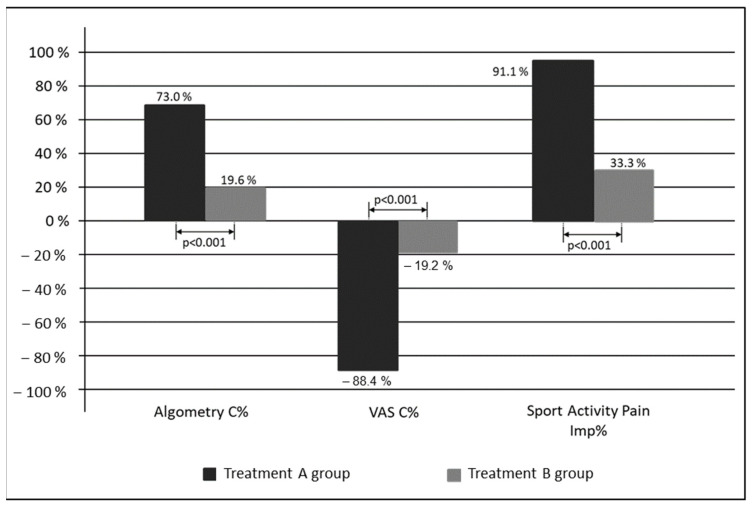
Change and improvement comparison between treatment A and B groups. C% = change %; Imp% = improvement %.

**Table 1 children-08-00963-t001:** Baseline characteristics of the children with calcaneal apophysitis.

Characteristic	TA (*n* = 104)	TB (*n* = 104)	ICC	*p* Value
Age (years.month)	11.1 ± 1.0	11.3 ± 1.0		0.133 ^1^
Male [*n* (%)]	85 (82.2)	88 (84.4)		0.842 ^2^
BMI (kg/m^2^)	19.2 ± 2.3	19.4 ± 2.4		0.677 ^1^
FPI-6 right	8.3 ± 1.7	8.2 ± 1.7		0.477 ^3^
FPI-6 left	8.3 ± 1.7	8.2 ± 1.7		0.580 ^3^
Lunge right (degrees)	32.3 ± 3.5	32.1 ± 3.7		0.761 ^3^
Lunge left (degrees)	32.4 ± 3.5	32.0 ± 3.7		0.447 ^3^
VAS (mm)	80.1 ± 13.1	81.3 ± 13.2		0.559 ^3^
Algometry (kgf)	2.9 ± 0.4	2.7 ± 0.4	0.91 [0.88–0.94]	0.026 ^3^

TA = treatment A group; TB = treatment B group; BMI = body mass index; FPI-6 = 6-Item Foot Posture Index; VAS = Visual Analogue Scale; ICC = Intraclass Correlation Coefficient. ^1^ Student’s t-test; ^2^ Fisher test; ^3^ Mann–Whitney U test. Qualitative variables are expressed as *n* (%), and quantitative variables as mean ± SD. The significance level was considered as *p* < 0.05.

**Table 2 children-08-00963-t002:** Outcomes at final assessment.

Outcome	Groups								β	Odds Ratio (95% CI) *
	TA (*n* = 100)				TB (*n* = 99)					
	Final	C %	Imp%	*p*-value	Final	C%	Imp%	*p*-value		
VAS (mm)	11.6 ± 17.4	−88.4 ± 18.5		<0.001 ^1^	67.3 ± 21.2	−19.2 ± 22.2		<0.001 ^1^	−68.6	(−74.5 to −62.7)
Algometry (kgf)	4.9 ± 0.5	73.0 ± 23.6		<0.001 ^1^	3.3 ± 0.7	19.6 ± 18.7		<0.001 ^1^	53.4	(47.1 to 59.7)

^1^ Mann–Whitney U test. * Treatment A group as reference. Change (C %) and improvement (Imp%) are calculated compared with baseline (Table 1). Qualitative variables are expressed as *n* (%), and quantitative variables as mean ± SD. TA = treatment A group; TB = treatment B group; VAS = Visual Analogue Scale. Beta estimates and corresponding 95% confidence intervals (95% CI). The significance level was considered as *p* < 0.05.

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
