# Peer review of "Effectiveness of Custom-Made Foot Orthoses vs. Heel-Lifts in Children with Calcaneal Apophysitis (Sever’s Disease): A CONSORT-Compliant Randomized Trial"

_children, 2021, doi:10.3390/children8110963_

Round 1

Reviewer 1 Report

Dear authors, thank you for your work. The paper regard about two different approaches in calcaneal apophysites. It is a well-constructed comparison study with clear criteria and valid evaluation tests.
The statistical analysis is rigorous and the results very interesting. In my opinion it is a well-executed study with interesting and useful conclusions for clinical practice.

Author Response

Dear reviewer, 

Thank you very much for your comments. We are glad that you apreciate our work.

We have used the MDPI English Editing Service in order to improve the english language of the manuscript.

Please see the attachment with is the english certificate. 

Reviewer 2 Report

Effectiveness of custom-made foot orthoses vs heel lifts in children with Calcaneal Apophysitis (Sever’s disease)

I would like to thank you for the opportunity to review this interesting paper. The question is relevant to the field and educational.

To my opinion, there is not a lot to be improved. The planning and concept of the study was thorough and without flaws. There are only some minor spelling mistakes, so I recommend to publish the paper after corrections.

Author Response

(The authors gave the same response as above.)
